# VariViT: A Vision Transformer for Variable Image Sizes

**Aswathi Varma**[1,2]                                          ASWATHI@TUM.DE
**Suprosanna Shit**[2,3]                                   SUPROSANNA.SHIT@TUM.DE
**Chinmay Prabhakar**[3]                               CHINMAY.PRABHAKAR@UZH.CH
**Daniel Scholz**[1,2]                                    DANIEL.SCHOLZ@MRI.TUM.DE
**Hongwei Bran Li**[4]                                      HOLI2@MGH.HARVARD.EDU
**Bjoern Menze**[3]                                          BJOERN.MENZE@UZH.CH
**Daniel Rueckert**[*2]                                    DANIEL.RUECKERT@TUM.DE
**Benedikt Wiestler**[*1]                                        B.WIESTLER@TUM.DE

[1] *Department of Neuroradiology, Technical University of Munich*

[2] *Institute for Artificial Intelligence and Informatics in Medicine, Technical University of Munich*

[3] *Department of Quantitative Biomedicine, University of Zurich*

[4] *Athinoula A. Martinos Center for Biomedical Imaging, Harvard Medical School, Boston*

**Editors:** Accepted for publication at MIDL 2024

## Abstract

Vision Transformers (*ViTs*) have emerged as the state-of-the-art architecture in representation learning, leveraging self-attention mechanisms to excel in various tasks. *ViTs* split images into fixed-size patches, constraining them to a predefined size and necessitating pre-processing steps like resizing, padding, or cropping. This poses challenges in medical imaging, particularly with irregularly shaped structures like tumors. A fixed bounding box crop size produces input images with highly variable foreground-to-background ratios. Resizing medical images can degrade information and introduce artefacts, impacting diagnosis. Hence, tailoring variable-sized crops to regions of interest can enhance feature representation capabilities. Moreover, large images are computationally expensive, and smaller sizes risk information loss, presenting a computation-accuracy tradeoff. We propose *VariViT*, an improved *ViT* model crafted to handle variable image sizes while maintaining a consistent patch size. *VariViT* employs a novel positional embedding resizing scheme for a variable number of patches. We also implement a new batching strategy within *VariViT* to reduce computational complexity, resulting in faster training and inference times. In our evaluations on two 3D brain MRI datasets, *VariViT* surpasses vanilla *ViTs* and *ResNet* in glioma genotype prediction and brain tumor classification. It achieves F1-scores of 75.5% and 76.3%, respectively, learning more discriminative features. Our proposed batching strategy reduces computation time by up to 30% compared to conventional architectures. These findings underscore the efficacy of *VariViT* in image representation learning.

**Keywords:** Vision Transformers, Architecture, Representation, Tumor Classification

## 1. Introduction

Deep neural architectures, notably Convolutional Neural Networks (CNNs) and Vision Transformers (*ViTs*), have emerged as effective architectures for image representation learning, consistently achieving state-of-the-art performance on real-world data across different

---

[*] Contributed equally as senior authors

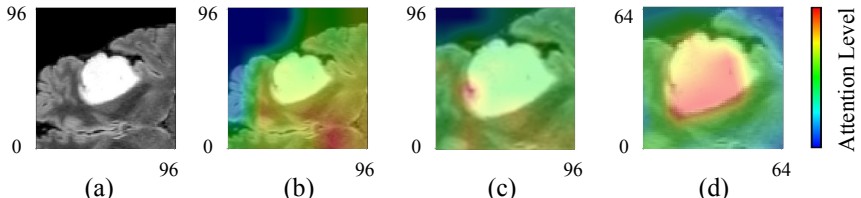

Figure 1: Selecting the optimal input crop size is essential for maximizing the representation quality (i.e., attention level) of *ViTs*. **(a):** 2D slice of the large tumor bounding box crop. **(b):** Attention map of the vanilla *ViT* model on the image, showing attention to background data rather than the tumor. **(c):** The image resized to a large fixed size shifts focus to distortions arising from the operation. **(d):** (Ours) Smaller image crop without resizing. In our method, attention is mainly given to desired tumor regions.

domains. *ResNet*, a CNN-based variant (He et al., 2016), achieves its efficacy in representation learning by the use of residual connections. *ViT* (Dosovitskiy et al., 2020) captures long-range dependencies by directly attending to global image information through self-attention mechanisms. *ViTs* are increasingly attracting attention in medical image analysis (He et al., 2021; Gao et al., 2021a; Chen et al., 2021; Shamshad et al., 2023; Gao et al., 2021b; Jang and Hwang, 2022). Input images are typically resized to a fixed size before being fed into a *ViT* model. These models perform well with evenly dispersed signals, allowing operations like interpolation and cropping for fixed-size inputs. However, medical images feature small, irregular regions of interest, where such methods can be detrimental. Fixed-size inputs result in varying foreground-to-background ratios, especially with smaller pathologies, where more background (e.g., healthy brain) is included compared to larger tumors. Moreover, medical images are particularly sensitive to distortions. Resizing them may introduce artificial features or modify existing ones, mimicking or obscuring real abnormalities and interfering with diagnosis. Figure 1 illustrates the attention map of a traditional *ViT* for tumor classification. Despite tumor regions, background areas are heavily attended to, wasting computational resources. Smaller crops resized with interpolation introduce distortions, causing unwanted focus shifts. These limitations might impair efficient model training.

We propose *VariViT* to handle variable-size images, addressing the limitation of fixed-size inputs. Our method recognizes the heterogeneous nature of real-world medical images where foreground-to-background ratios vary significantly. *VariViT* retains the favorable properties of *ViTs* while integrating the capability to handle diverse image sizes. Our contributions are as follows:

1. We introduce a novel **flexible positional embedding strategy** tailored to different image sizes.
2. We propose an **alternate batching strategy** to improve computational efficiency, leveraging on the inclusion of smaller-sized images.
3. We demonstrate the **applicability of *VariViT*** in (i) glioma genotype prediction and (ii) brain tumor classification, two challenging tasks due to the highly variable tumor sizes. Our extensive experiments on two brain MRI datasets highlight the superior performance of *VariViT* over both vanilla *ViT* and ResNet architectures.

## 2. Related Work

The traditional *ViT* model uses a fixed input size. It either initializes fixed positional embeddings or learns them during training. These embeddings are linearly interpolated for fine-tuning and evaluation at higher resolutions. The *FlexiViT* model (Beyer et al., 2023) also adapts to variable image-to-patch size ratios and sequence lengths by resizing the learnable 2D positional embedding grid with bilinear interpolation. While effective in 2D, interpolating the 3D embedding grid for variable-sized images may result in information loss and higher computational requirements due to more complex calculations. The *SuperViT* model (Lin et al., 2023) patchifies an image at multiple scales and improves computational cost by randomly dropping tokens. However, a random selection of tokens risks information loss and degrades representation quality, particularly in medical images where the region of interest may be small.

The *Pix2Struct* model (Lee et al., 2023), similar to ours, handles variable image sizes. The vision encoder resizes the input images to extract fixed-size patches fitting a predefined sequence length. Padding is applied to the sequence as needed, allowing it to reach the desired fixed length. The model learns a large grid of 2D absolute positional embeddings, enabling the identification of patch positions based on $x$ and $y$ coordinates. However, this approach can be computationally expensive, especially for 3D images. In contrast, our model efficiently manages size differences without resorting to such computationally expensive operations. Moreover, scaling the images can introduce undesired artefacts.

The *NaViT* model (Dehghani et al., 2023) also addresses challenges on computational complexity and variable image sizes. *NaViT* packs patches from different-sized images into the same sequence. It maintains a fixed sequence length by randomly dropping tokens and padding. The model employs masked attention and pooling to prevent interactions between patches from different images. However, dropping tokens may lead to information loss, and implementing masked attention and pooling introduces more complex architectural changes.

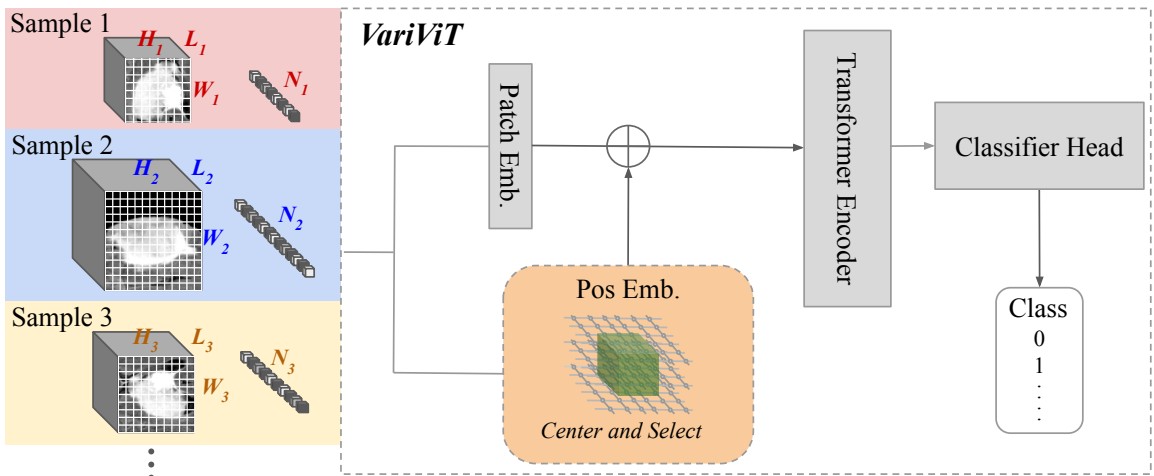

Figure 2: The *VariViT* model addresses the problem of handling images with different sizes by introducing a novel positional embedding resizing mechanism and employing different batching strategies. The model utilizes a fixed patch size, ensuring consistent patch embedding sizes across images while simultaneously adapting to different sequence lengths using a *center and select* resizing strategy.

## 3. Method

**Overview.** We focus on learning 3D image representation in this work. In the conventional *ViT* framework, a 3D input image is divided into non-overlapping patches. These patches are flattened and linearly projected to obtain a sequence of patch embeddings. The sequence is fed into the transformer encoder after adding a CLS token. The enriched CLS token is used for the final classification. However, the transformer model lacks intrinsic awareness of the spatial arrangement of patches within the sequence. Therefore, positional embeddings are added to the sequence to explicitly incorporate this information. Both *relative* and *absolute* positional encoding can be employed (Vaswani et al., 2017; Wu et al., 2021). They can either be *fixed* or *learned* during training. In its simplest form, absolute fixed *sinusoidal* embeddings are used for this purpose.

Fixed positional embeddings are predefined vectors representing each patch's absolute coordinates within the input sequence, typically generated using sinusoidal functions. The 1D positional encoding for even and odd indices is formulated as (Vaswani et al., 2017):

$$\begin{aligned}
\text{PE}(pos, 2i) &= \sin\left(pos/10000^{2i/d}\right) \\
\text{PE}(pos, 2i+1) &= \cos\left(pos/10000^{(2i+1)/d}\right)
\end{aligned} \tag{1}$$

*pos* represents the position to be encoded. The parameter $1/10000^{2i/\text{d}}$ governs the wavelength of the sinusoids. Here, $d$ denotes the embedding dimension and $i$ refers to each of the individual dimensions of the embedding.

Positional embeddings can be adjusted for various image sizes through interpolation, as suggested in the *ViT* paper (Dosovitskiy et al., 2020). However, interpolation increases computational complexity and may introduce approximations, making it suboptimal for resizing 3D embedding grids in variable-sized tumor crops. *Pix2Struct* suggests learning a large grid of embeddings for a predefined sequence length (Lee et al., 2023). This, however, can extend training time. Leveraging the consistent center alignment in tumor crops can provide a reliable reference point for resizing positional embeddings without the need for interpolation, thus preserving the original information.

Building upon this concept, the *VariViT* model (Figure 2) introduces the *center and select* method for resizing positional embeddings. Our model adapts the vanilla *ViT* architecture for various input image sizes, particularly tailored to heterogeneous tumor shapes, while maintaining a fixed patch size. This is achieved by extracting 3D bounding box crops categorized into three sizes by tumor volume. Additionally, we exploring batching methods to manage diverse image sizes within batches. The architectural and training modifications to the base *ViT* are explained in the following sections.

**Center and Select**. Similar to the vanilla *ViT*, the patch embedding size remains constant in *VariViT* regardless of the image size. However, with a fixed patch size, variations in image size can lead to a difference in the number of patches or the sequence length. Consequently, positional embeddings must be dynamically resized to accommodate these variations. In tumor classification tasks, 3D crops are often employed to isolate the tumor region by identifying tumor boundaries using segmentation masks. The center coordinates of the 3D crop are aligned with the center of mass of the tumor. Despite variations in size, tumors

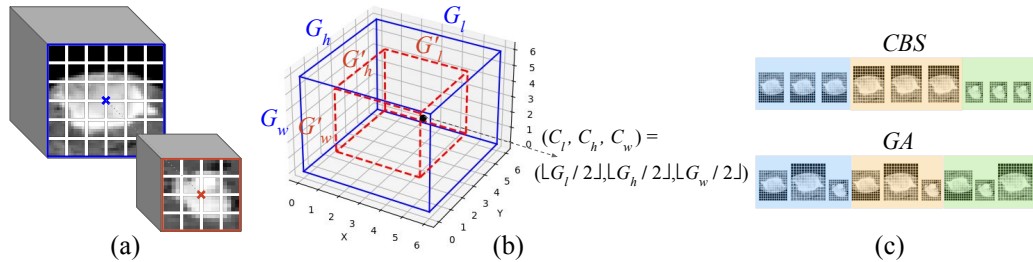

Figure 3: **(a):** Tumor bounding boxes of two sizes are displayed, with the tumor center of mass aligned. **(b):** Our proposed *center and select* method for resizing position embeddings initializes fixed sinusoidal embeddings for the largest image size. Embeddings for other sizes are selected from the center of this grid. **(c):** We present two batching strategies: *Custom Batch Sampler (CBS)* with the same image sizes and *Gradient Accumulation (GA)* with varying image sizes within a batch.

positioned at the centers of the 3D crops will have coinciding centers of mass, establishing a consistent reference point (Figure 3 - (a)). Leveraging this, we introduce the *center and select* method, resizing positional embeddings by centering them in 3D space.

The implementation involves initializing a fixed positional embedding for the largest image size. We utilize sinusoidal encoding as described by Equation 1 extended to 3D coordinates. This results in positional embeddings of dimension $[N, d]$, excluding the CLS token. Here, $N$ is the number of patches and $d$ is the embedding dimension. The embedding can be viewed as a 3D grid for the $l$, $h$, and $w$ dimensions with a size of $G_l \times G_h \times G_w = N$ (Figure 3 - (b)). The center of the grid $(C_l, C_h, C_w)$, determined as $\left( \left\lfloor \frac{G_l}{2} \right\rfloor, \left\lfloor \frac{G_h}{2} \right\rfloor, \left\lfloor \frac{G_w}{2} \right\rfloor \right)$, serves as the reference point for selecting a subset of positions based on the current input image size. We dynamically compute the new positional embedding size $[N', d]$ when the image dimensions differ from the largest size, resulting in a new grid size $G'_l \times G'_h \times G'_w$. To adjust the positional embedding for a different image size, we select a subset around the center from the initialized positional embedding. The position range $[start, end)$ for each dimension is determined by $start = C_k - \left\lfloor \frac{G'_k}{2} \right\rfloor$ and $end = start + G'_k$, where $k = l, h, w$. Thus, the original positional information is extracted at a lower computational cost.

**Batching Methods.** Training models with different input dimensions poses challenges due to the variability of bounding box sizes in the dataset. Existing Python frameworks lack seamless solutions to address this issue. To address this challenge, we opt for two specific strategies, namely a *custom batch sampler* and *gradient accumulation*.

1. **Custom Batch Sampler (CBS)** - This strategy involves grouping images of the same size into a batch, as shown in Figure 3 (c) while allowing the image size to vary randomly from batch to batch. This maintains consistency within each batch. Including batches with smaller image sizes contributes to significantly faster training.

2. **Gradient Accumulation (GA)** - In this method, the weight update is performed after accumulating gradients over several mini-batches, resulting in batches with varying image sizes (Figure 3 - (c)). The effective batch size is given by: *Batch Size = Mini-Batch Size × Update Interval*. Here, we adjust the mini-batch size to 1 and set the update interval to the desired batch size.

We make our codes for *VariViT* and the batching schemes publicly available at https://github.com/Aswathi-Varma/varivit.

## 4. Experimental Setup

**Datasets.** To highlight the effectiveness of *VariViT*, we perform experiments on two distinct 3D brain MRI datasets (Appendix A): The *glioma* dataset comprising 1856 MRI scans sourced from various studies (van der Voort et al., 2021; Sayah et al., 2022; Calabrese et al., 2022; Bakas et al., 2022, 2017); and the *brain tumor* dataset containing 1699 MRIs from publicly available datasets (Baid et al., 2021; Suter et al., 2022; Moawad et al., 2023). Both datasets contain FLAIR, T2w, T1w, and T1w+contrast MR images, all registered to the SRI24 atlas and resampled to a uniform voxel size of 1x1x1 mm³, forming a four-channel multi-modal input for our experiments. These datasets are chosen to evaluate the model's performance on two binary classification tasks: (i) identifying the *isocitrate dehydrogenase (IDH)* mutation status, a key biomarker that separates two adult-type diffuse gliomas groups (Louis et al., 2021). (ii) distinguishing between primary brain tumors (gliomas) and secondary brain tumors (metastases). We use the glioma dataset for multi-class classification task, targeting three glioma subtypes: *glioblastoma*, *astrocytoma*, and *oligodendroglioma*.

We extract the largest tumor in each patient for our baseline models by cropping a 96×96×96 mm³ bounding box guided by segmentation masks provided in the datasets. To address different tumor sizes in *VariViT*, we categorize the datasets into three size bins with equal sample distribution (Appendix B). These bins correspond to crop sizes of 64×64×64 mm³, 80×80×80 mm³, and 96×96×96 mm³ for the largest tumors. Additionally, we rescale all image intensities to the range [0, 1].

**Training and Evaluation.** In our training configuration, we opt for the 3D *ViT* model (Prabhakar et al., 2023). We employ the *ViT*-S/16 configuration with a patch size of 16. This setup consists of 12 encoder blocks, each having an embedding dimension of 384 and 6 attention heads. A linear layer is used as the classification head. For both the *VariViT*-S/16 models (GA and CBS), we use the same vanilla *ViT* base, differing only in the positional embedding. The model comprises approximately 28 million trainable parameters. We utilize *ResNet-18*, a CNN with 33 million parameters to compare our results with a convolutional model of roughly the same number of parameters. To benchmark against recent state-of-the-art variable image size models, we incorporate the *Pix2Struct* vision encoder into a 3D framework. This model serves as one of our baselines, with 34.8 million trainable parameters.We fix the sequence length of *Pix2Struct* at 216, corresponding to our dataset's maximum image size. For all the models, we utilize absolute positional encoding. We explore relative positional embedding with our batching methods in Appendix C but do not observe any significant advantage over the absolute approach.

All models undergo training for 100 epochs, utilizing the AdamW optimizer (Loshchilov and Hutter, 2017) with a weight decay of 0.05. The base learning rate is set to 1e-3, and cosine decay (Loshchilov and Hutter, 2016) is applied for learning rate decay. A batch size of 8 is used, and the experiments are conducted on Nvidia RTX A6000. A warm-up schedule (Goyal et al., 2017) of 40 epochs is applied. Data augmentations such as *random affine*, *random noise*, *random gamma*, *random blur*, and *random flips* are incorporated during training. Cross-entropy loss with class weights is employed to mitigate class imbalance in the dataset. For the evaluation metrics, we utilize the *Area Under the Curve* (AUC), *F1-score*, and *Matthews Correlation Coefficient* (MCC) score.

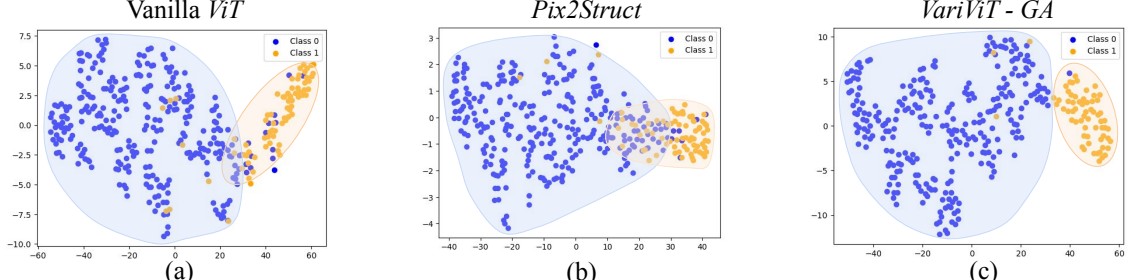

Figure 4: t-SNE visualization of embedding layer output for *IDH* status classification. **(a)**: Vanilla *ViT*, **(b)**: *Pix2Struct* and **(c)**: *VariViT-GA*, all with an embedding dimension of 384. Notice the clearer separation of clusters in our model's plot.

## 5. Results

**IDH Mutation Status.** We train our proposed model on the glioma dataset for *IDH* status classification task. Then, we compare its performance against three baseline models: fixed-size 3D *ResNet-18*, 3D vanilla *ViT*, and variable-size 3D *Pix2Struct*. We employ a k-fold cross-validation strategy with k=5 for all the models and report the mean metrics and standard deviation obtained from the test sets. The *VariViT-GA* model consistently outperforms the baseline models in terms of AUC, F1-score, and MCC (Table 1). Our model also exhibits notably faster training times compared to its counterparts. *VariViT-CBS* not only outperforms *ResNet-18* and vanilla *ViT* on various performance metrics, including AUC, but also further reduces the training time. The t-SNE plot (Van der Maaten and Hinton, 2008) in Figure 4 illustrates the improved cluster separation achieved by our model.

|  | Method | AUC | F1-Score | MCC | Training Time |
|---|---|---|---|---|---|
| *Fixed* | 3D ResNet-18 | $0.928 \pm 0.042$ | $0.716 \pm 0.058$ | $0.654 \pm 0.071$ | 1.783 |
|  | 3D Vanilla-ViT | $0.927 \pm 0.027$ | $0.744 \pm 0.059$ | $0.679 \pm 0.076$ | 1.785 |
| *Variable* | 3D Pix2Struct | $0.940 \pm 0.012$ | $0.742 \pm 0.040$ | $0.686 \pm 0.056$ | 1.631 |
|  | VariViT-CBS | $0.937 \pm 0.009$ | $0.718 \pm 0.027$ | $0.653 \pm 0.028$ | 1.292 |
|  | VariViT-GA | $\mathbf{0.942} \pm 0.011$ | $\mathbf{0.755} \pm 0.059$ | $\mathbf{0.709} \pm 0.069$ | 1.502 |

Table 1: Comparison of *VariViT* with baseline models for the *IDH* mutation status prediction task. Average training times visualized on the right (hours).

**Brain Tumor Type.** To further highlight the effectiveness of our proposed model, we apply it to the brain tumor dataset for the classification of primary versus metastatic tumors. Comparing the *VariViT-GA* model with the baselines, it distinctly outperforms *Pix2Struct* and *ResNet-18*. Our model shows superior performance in MCC and F1-scores compared to Vanilla *ViT* (Table 2). The *VariViT-CBS* surpasses *Pix2Struct* in performance and achieves similar results to *ResNet-18*, with faster training times. This underscores its efficiency and suitability for practical applications.

**Ablation Study - Positional Embedding.** Here, we compare the effectiveness of different positional embedding methods within the *VariViT-GA* architecture. We analyze three position-embedding strategies alongside the center and select method: (i) *Indepen-*

| | Method | AUC | F1-Score | MCC | Training Time |
|---|---|---|---|---|---|
| *Fixed* | 3D ResNet-18 | $0.948 \pm 0.013$ | $0.745 \pm 0.035$ | $0.694 \pm 0.051$ | 1.804 |
| | 3D Vanilla-ViT | $\mathbf{0.957} \pm 0.011$ | $0.752 \pm 0.067$ | $0.696 \pm 0.081$ | 1.720 |
| *Variable* | 3D Pix2Struct | $0.945 \pm 0.020$ | $0.720 \pm 0.058$ | $0.663 \pm 0.073$ | 1.620 |
| | VariViT-CBS | $0.947 \pm 0.013$ | $0.746 \pm 0.035$ | $0.686 \pm 0.045$ | 1.200 |
| | VariViT-GA | $0.954 \pm 0.007$ | $\mathbf{0.763} \pm 0.036$ | $\mathbf{0.706} \pm 0.046$ | 1.507 |

Table 2: Comparison of *VariViT* with baseline models for the primary *vs.* secondary brain tumor classification. Average training times are visualized on the right (hours).

*dent, Fixed* (Indep_Fixed) - initializes separate fixed sinusoidal positional embeddings for each image size category. (ii) *Interpolated, Fixed* (Interp_Fixed) - initializes fixed sinusoidal embedding for the largest image size and employs trilinear interpolation for smaller image sizes. (iii) *Interpolated, Learned* (Interp_Learned) - uses the positional embedding learned from the largest image size to create embeddings for smaller images through trilinear interpolation. All methods perform effectively (Table 3), but the *center and select* approach produces better results for the *IDH* status classification task.

**Ablation Study - Multi-Class Classification.** We extend the glioma dataset to classify three glioma subtypes, thereby evaluating the model's performance in this more complex task. Our model achieves comparable MCC scores to both *ResNet-18* and *Pix2Struct* (Table 4), while demonstrating faster training times. This study underscores the effectiveness of our model across diverse classification tasks, especially in scenarios involving multiple classes.

| Method | AUC | F1-Score | MCC |
|---|---|---|---|
| Indep_Fixed | $0.938 \pm 0.011$ | $0.742 \pm 0.076$ | $0.701 \pm 0.074$ |
| Interp_Fixed | $0.929 \pm 0.007$ | $0.720 \pm 0.065$ | $0.677 \pm 0.048$ |
| Interp_Learned | $0.940 \pm 0.008$ | $0.750 \pm 0.025$ | $0.690 \pm 0.034$ |
| Center & Select | $\mathbf{0.942} \pm 0.011$ | $\mathbf{0.755} \pm 0.059$ | $\mathbf{0.709} \pm 0.069$ |

| Method | MCC |
|---|---|
| 3D ResNet-18 | $\mathbf{0.548} \pm 0.04$ |
| 3D Vanilla ViT | $0.519 \pm 0.07$ |
| 3D Pix2Struct | $0.543 \pm 0.02$ |
| VariViT-GA | $0.544 \pm 0.06$ |

Table 3: Comparison of positional embedding strategies using the *VariViT-GA* model.

Table 4: Comparison of models for multi-class classification.

## 6. Discussion and Conclusion

*ViTs* excel at image feature learning, but limitations exist in medical image analysis due to computational burden and anatomical variability. We address this with *VariViT*, a method that efficiently scales to various 3D image sizes while effectively learning for improved classification. Our approach centers around maintaining focus on the region of interest, a strategy that demonstrably improves feature learning. By adapting to variable image sizes, *VariViT* concentrates on critical areas despite inherent anatomical and/or pathological variability. This targeted approach, however, necessitates an initial bounding box or segmentation. While we tested our framework on brain MRIs, its versatility allows for adaptation to other modalities and regions, and we encourage to adapt our method to individual needs. A significant avenue for future research is the exploration of our batching strategy and positional embedding technique in the context of extremely large datasets or high-resolution images. In such scenarios, we anticipate that the efficiency gains of our method will be particularly advantageous when compared to traditional *ViT* implementations.

## Acknowledgments

This study was supported by the DFG within the SPP Radiomics, grant #428223038.

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

## Appendix A. Datasets

The *glioma* dataset comprises MRIs from various studies, including the *Erasmus Glioma Database* (EGD) (van der Voort et al., 2021), the *REMBRANDT* MRI dataset collection (Sayah et al., 2022), the *University of California San Francisco Preoperative Diffuse Glioma MRI Dataset* (UCSF-PDGM) dataset (Calabrese et al., 2022), the *University of Pennsylvania Glioblastoma Imaging, Genomics, and Radiomics* (UPenn-GBM) dataset (Bakas et al., 2022), *The Cancer Genome Atlas* (TCGA) (Bakas et al., 2017), and a private MRI dataset. In the *brain tumor* dataset, we collect data from the *BraTS 21* (Baid et al., 2021) and *LUMIERE* (Suter et al., 2022) datasets for primary tumors. For metastases, we utilize the *BraTS-Mets 2023* (Moawad et al., 2023) dataset.

## Appendix B. Bounding Box Distribution

To simulate various image sizes, we categorize both datasets into three bins, each containing approximately equal numbers of samples, based on the dimension of the largest tumor.

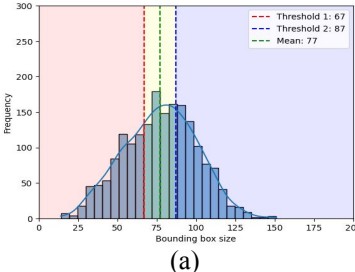 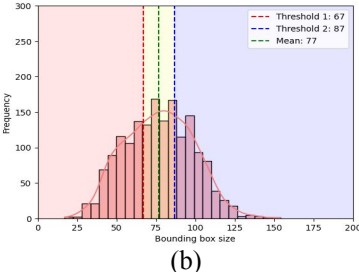

(a)          (b)

Figure 5: 3D Bounding Box Distribution of Glioma (a) and Brain Tumor (b) datasets. The x-axis represents the size of the 3D bounding box along the three dimensions, while the y-axis denotes the frequency of samples. The distribution of samples is divided into three equal bins based on the size of the bounding boxes.

We establish two threshold values to ensure an equal distribution of samples across these bins. Subsequently, a fixed bounding box crop size is assigned for each bin: 64x64x64 mm³ for cases where the threshold is less than 67, 80x80x80 mm³ for threshold values between 67 and 87 (inclusive), and 96x96x96 mm³ for the largest size. This is depicted in Figure 5 showcasing the tumor size distribution for the glioma (left) and brain tumor (right) datasets. In the brain tumor dataset, metastatic samples with tumor bounding box sizes smaller than 40x40x40 mm³ are excluded to ensure adequately sized tumors, thereby enhancing the complexity of the classification task.

## Appendix C. Absolute v/s Relative Position Embedding

*Absolute* positional encoding techniques allocate distinct encoding vectors to every position within the input sequence, thereby allowing the model to capture the absolute positions up to the maximum sequence length. These methods employ either fixed or learnable encodings. In contrast, *relative* position methods encode the relative distance between input patches and learn the pairwise relationship between them (Shaw et al., 2018; Wu et al., 2021). Typically, this is computed through a look-up table with learnable parameters that interact with queries and keys within self-attention modules during the training process. We

| Batching | Coordinates | AUC | F1-Score | MCC |
|----------|-------------|-----|----------|-----|
| *CBS* | Absolute | **0.933** ± 0.013 | **0.712** ± 0.032 | **0.646** ± 0.036 |
|  | Relative | 0.933 ± 0.009 | 0.697 ± 0.013 | 0.622 ± 0.019 |
| *GA* | Absolute | **0.945** ± 0.007 | **0.744** ± 0.036 | **0.684** ± 0.034 |
|  | Relative | 0.931 ± 0.016 | 0.718 ± 0.033 | 0.666 ± 0.031 |

Table 5: Comparison of absolute and relative position embeddings for both the batching methods on the glioma dataset. Note that the embeddings are learned, and resizing is done by interpolation.

experiment with both relative and absolute positional embeddings using our two proposed batching strategies for variable image sizes on the glioma dataset. For both methods, we initialize the positional embedding with the dimensions of the largest image size. We employ interpolation to adjust its size when dealing with varying image dimensions. *CBS* and *GA* batching with absolute positional embedding demonstrate superiority over their relative

counterparts. Hence, relative positional embedding doesn't offer a significant advantage over absolute for our batching methods.

## Appendix D. Cosine Similarity

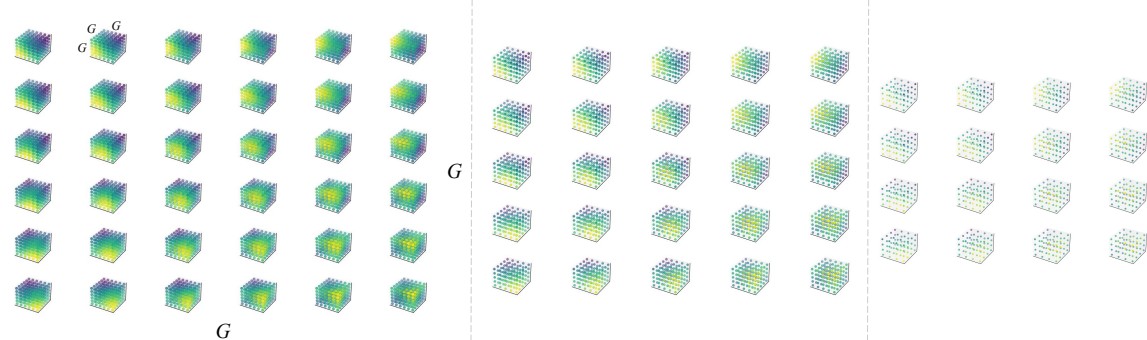

Figure 6: 2D view of the cosine similarity visualization of *VariViT*'s positional embedding for each of the three image sizes.

Cosine similarity quantifies the similarity between two vectors in a $d$-dimensional space, determined by the cosine of the angle between them. Values range from 0 to 1, where 1 signifies perfect similarity. In Figure 6, each cube depicts similarity between one position's embedding and the remaining $N - 1$ positions, where $N = G \times G \times G$ denotes the total elements in the position embedding grid.

## Appendix E. t-SNE Plots

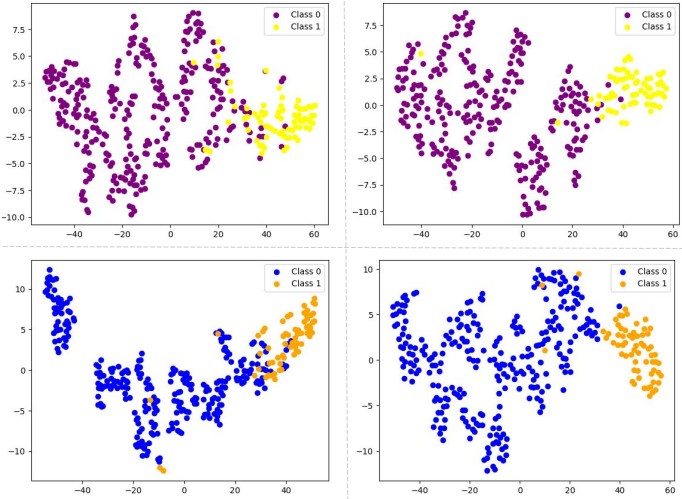

Figure 7: t-SNE Plots depicting *VariViT* CBS on the left and GA on the right, showcasing evaluations based on the highest scores across k-folds for Glioma (top) and Brain tumor (bottom) datasets.

