# OpenReview forum: "VariViT: A Vision Transformer for Variable Image Sizes"
_MIDL.io/2024/Conference — MIDL 2024 Poster_

### Official Review · Reviewer_oTEG · 2024-02-20

**Confidence:** 4
**Preliminary Rating:** 3
**Final Rating:** 4

**Summary:**

Most of the current Vision Transformer (ViT) architectures rely on fixed-size images which implies preprocessing steps on images like resizing, cropping, and padding. For example, a fixed crop size produces input images with highly variable foreground-to-background ratios while resizing medical images may degrade information and introduce artifacts. This may impact the robustness of the model. To handle variable-sized images, this paper adapts the original ViT architecture proposing more flexible absolute positional embeddings and a batching strategy that groups images of the same size into the same batch. The method is evaluated on two 3D brain MRI datasets.

**Strengths:**

- Handling variable-sized images with ViT is an interesting problem and useful for the community.
- The paper is well written, and the problem and the solution are clear.
- The two proposed modifications (flexible position embedding and alternate batching strategy) are simple and seem architecture agnostic. Therefore, they could be adaptable to other ViT architectures.

**Weaknesses:**

I like the fact that the method is simple and effective but I have concerns about the comparisons with related works that also propose ViT architecture that can handle variable-sized images. How does the proposed VariViT perform compared to these architectures like Pix2Struct or NaViT? Authors said that because these architectures are dropping tokens it may lead to information loss and hence worse representation quality. However, no quantitative evaluations allow them to conclude about this and the superiority of VariViT compared to Pix2Struct or NaViT. That is why, in addition to the baseline (3D ResNet-18, 3D Vanilla ViT), it would be interesting to see how the method compares to these concurrent works.

**Detailed Comments:**

- This point is more a discussion/suggestion rather than a weakness. In some medical imaging tasks like MRI classification (3D) or histopathology slide classification (2D), the relative position information of  3D patches (MRI) or 2D patches (histopathology) may be more relevant and useful for the task compared to the absolute positional information. Additionally, relative positional information as it is done for instance in [1, 2] is usually injected at each self-attention block which may help the learning and they can also extrapolate at some point to image sizes larger than the ones seen during training. That is why, I was wondering if the authors tried such relative position encoding and if this combined with your custom batch sampler could work.
- If I understood correctly, your absolute positional embeddings are built based on the largest image size in the training dataset. So I was wondering what would happen if a test image has a size larger than the largest one of the training dataset?

References:
- [1] Su, Jianlin, et al. "Roformer: Enhanced transformer with rotary position embedding." Neurocomputing 568 (2024): 127063.
- [2] Press, Ofir, Noah A. Smith, and Mike Lewis. "Train short, test long: Attention with linear biases enables input length extrapolation." arXiv preprint arXiv:2108.12409 (2021).

**Justification Of Final Rating:**

The authors have adressed most of my concerns and answered some questions I had.
I particularly appreciate the effort of providing comparisons with previous work while extending their work to the 3D data case. Therefore, I am increasing my grade to weak accept.

**Justification Of The Preliminary Rating:**

In the current form, since there are no comparisons with related works, we cannot conclude about the utility/superiority of this work compared to existing methods. If the authors address my concerns, I would be happy to increase my grade.

**Questions To Address In The Rebuttal:**

I would like the authors to address my concerns in the Weaknesses section but also discuss the points raised in the Detailed Comments section.

**Special Issue:**

No

---

> ### Author Response · Authors · 2024-03-17
>
> We thank the reviewer for their thoughtful evaluation of our paper on *VariViT*. We appreciate your recognition of the strengths of our work, particularly in handling variable-sized images with *ViT*, and the clarity of our problem statement and proposed solution. We value your feedback regarding potential weaknesses and suggestions for improvement. We have taken your input into consideration and made the necessary adjustments accordingly.
>
> ### 1. Comparison with Related Works
>
> Regarding the comparison with related works such as *Pix2Struct* and *NaViT*, we agree that it is essential to provide a comprehensive evaluation to demonstrate the superiority of *VariViT*. In response to your comment, we conducted additional experiments to compare *VariViT* not only with vanilla *ViT* and ResNet but also with *Pix2Struct*. Despite *Pix2Struct's* initial operation in a 2D domain, we extended its application to a 3D framework. Our model demonstrates better performance over *Pix2Struct*, achieving a **1.3%** improvement in the F1-score for *IDH* status classification and a **4.3%** improvement for tumor type classification. Notably, this is achieved with up to an **8%** reduction in training time. To ensure full reproducibility, we also release our 3D implementation of *Pix2Struct* along with the *VarViT* code publicly at [https://github.com/Aswathi-Varma/varivit](https://github.com/Aswathi-Varma/varivit). We were unable to include *NaViT* in our comparison due to two factors. First, the authors did not release their code publicly. Second, certain aspects of their paper lacked clarity, making it difficult to implement *NaViT* within the limited time available for rebuttal.
>
> ### 2. Experimentation with Relative Positional Encoding
>
> Regarding the reviewer's suggestion on relative positional encoding, we appreciate your insight into the potential relevance and usefulness of relative position information in certain medical imaging tasks. We experimented with relative positional embeddings alongside our custom batch sampling methods. Although we did not observe significant advantages over absolute positional embeddings in our experiments, we acknowledge the importance of relative position information within certain contexts. As such, we have included a comparison in the *Appendix. C* of the revised paper.
>
> ### 3. Handling Test Images Larger Than the Largest Training Image Size
>
> Indeed, our method relies on the assumption that all positions have been encountered during training, including those for the largest image size present in the dataset. We achieve this by utilizing segmentation masks to determine tumor sizes and dividing the dataset into predefined bins. Consequently, our train-test split ensures that both sets contain similar bin sizes, guaranteeing that all sizes in the test set have been encountered during training. If a test image exceeds the size of the largest one in the training dataset, our current model does not support such scenarios. While resizing by interpolation could address this issue, we've prioritized lower training and inference times without interpolation for efficiency.

---

> > ### Comment · Reviewer_oTEG · 2024-03-21
> >
> > The authors have adressed most of my concerns and answered some questions I had.
> > I particularly appreciate the effort of providing comparisons with previous work while extending their work to the 3D data case. Therefore, I am increasing my grade to weak accept.

---

### Official Review · Reviewer_jPbu · 2024-02-29

**Confidence:** 3
**Preliminary Rating:** 4
**Recommendation:** Poster
**Final Rating:** 5

**Summary:**

The key idea behind this paper is the development of VariViT, "an improved ViT model" designed to handle variable sized medical images. This paper includes:

1. "A flexible positional embedding strategy tailored to different image sizes"
2. "An alternate batching strategy"
3. Demonstration of VariViT on a couple of different medical imaging tasks.

**Strengths:**

The paper introduces VariViT, "an improved ViT model" designed to address the challenges of handling variable image sizes in medical imaging. This innovation is significant given the limitations of traditional ViTs and CNNs in dealing with the variability and sensitivity of medical images.

**Weaknesses:**

The paper focuses exclusively on brain MRI datasets for glioma genotype prediction and brain tumor classification. The generalizability of VariViT to other medical imaging modalities, body parts, and disease types remains unexplored. Extending the evaluation to include diverse datasets could provide a more comprehensive understanding of the model's applicability and robustness across different medical imaging challenges.

The paper presents an ablation study on positional embedding methods, which is valuable. However, a more comprehensive set of ablation studies, including the impact of different patch sizes, the effect of the novel batching strategy on different types of medical images, and variations in the model architecture, could offer deeper insights into the model's critical components and their contributions to its performance.

Although VariViT shows improved computational efficiency over traditional architectures, the scalability of the proposed batching strategy and positional embedding approach in extremely large datasets or very high-resolution images is not thoroughly discussed.

**Detailed Comments:**

Minor revisions for language and clarity could improve the paper's accessibility. This includes simplifying complex sentences and ensuring consistency in terminology.

**Justification Of Final Rating:**

The authors have addressed most of my concerns clearly and have provided proof of additional experiments. Authors have provided sufficient work for me to increase from my initial rating of weak accept.

**Justification Of The Preliminary Rating:**

The paper introduces an innovative solution tailored to tackle the issue presented. Despite its strengths, the manuscript falls slightly short in offering a comprehensive evaluation by not adequately addressing the model's limitations.

**Questions To Address In The Rebuttal:**

While the paper presents its strengths, a more explicit discussion on the limitations of VariViT, including potential biases, dataset limitations, and areas where the model might not perform as expected, would provide a balanced view and suggest avenues for future research.

Minor revisions for language and clarity could improve the paper's accessibility. This includes simplifying complex sentences and ensuring consistency in terminology.

**Special Issue:**

Yes

---

> ### Author Response · Authors · 2024-03-17
>
> Thank you to the reviewer for his/her insightful feedback on our paper. We are pleased to see that the reviewer acknowledges the significance of our contribution in introducing *VariViT*, an improved *ViT* model tailored to address the challenges posed by variable-sized medical images. We agree that traditional *ViTs* and *CNNs* often struggle with the variability and sensitivity inherent in medical imaging, and our innovation aims to bridge this gap.
>
> ### 1. Limited Dataset Diversity:
>
> Our research primarily focused on brain MRI datasets, influenced by the expertise of our **neuroradiology** research group. However, we recognize the significance of assessing *VariViT* across various medical imaging modalities. We expect that the adaptability and flexibility of *VariViT* make it easily extendable to pathologies of other body regions. Therefore, we plan to extend our evaluation to include a diverse range of medical imaging tasks involving different anatomical sites to provide a more comprehensive understanding of *VariViT's* applicability and robustness in various medical imaging challenges. We now include this aspect (as well as scalability, see below) in our discussion.
>
> ### 2. Scope of Ablation Studies:
>
> The reviewer pointed out that our ablation studies could be expanded to include a more comprehensive set of experiments. We agree that exploring aspects like patch sizes, the new batching strategy on various medical images, and variations in the model architecture would be valuable. We plan to conduct additional experiments to gain deeper insights into these critical components and their contributions to *VariViT's* performance. Additionally, we incorporate an analysis of the model's performance in multi-class classification tasks within our revised manuscript. We also explore *relative* positional encoding because of its relevance in certain medical imaging tasks. While integration with our batch sampling methods showed feasibility, no significant advantages over absolute positional embeddings were observed. We include a comparison of both methods in Appendix C of the revised paper.
>
> ### 3. Scalability Concerns:
>
> We appreciate the reviewers raising concerns about the scalability of our proposed batching strategy and positional embedding approach in extremely large datasets or high-resolution images. To maintain a focused initial study, we limited the scope of our experiments. However, we plan to conduct further investigations specifically in the context of handling large datasets and high-resolution images.

---

### Official Review · Reviewer_eAzb · 2024-02-29

**Confidence:** 4
**Preliminary Rating:** 4
**Recommendation:** Poster
**Final Rating:** 4

**Summary:**

This paper proposed a novel strategy of "center and select" for resizing positional embedding in Vision Transformer, handling variations in object size. The authors also proposed and tested two strategies of batching methods to address the challenge of different input dimensions. On two 3D brain MRI datasets, VariViT surpasses vanilla ViTs in genotype prediction and tumor classification tasks, with a clearer separation of clusters in tSNE visualization of embedding layer output.

**Strengths:**

The paper has its merit in addressing the challenge of irregularly shaped structures like tumors, which is of interest in the field of medical image computing. The "center and select" method is intuitive and effective for resizing position embeddings.

The proposed CBS strategy accelerates training and the proposed GA strategy handles varying image sizes within a batch and achieves the best performance.

The paper is clear, well-written, and easy to follow.

**Weaknesses:**

The proposed method seems to require segmentation masks as part of the dataset since it computes the largest size of tumor bounding boxes the first, which could be a potential limitation.

The two tasks involved in this paper are only binary classification tasks, i.e., outputs 0 and 1. It will be great to also extend the evaluation for multi-class tasks.

The authors only compared their proposed method with vanilla ViTs and Res-Net, and through ablation studies with fixed/learned and independent/interpolated positional embedding strategies, but it will also be of interest to include a comparison with other related works, especially the two similar works Pix2Struct and NaViT.

**Detailed Comments:**

Minor:

Figure 4: suggest adding the labels for each class of model instead of only putting it next to the table. The same suggestion goes to Figure 6 as well.

**Justification Of Final Rating:**

I appreciate that the authors addressed the concerns and made substantial changes to the paper. I am keeping the weak acceptance decision since the requirement of segmentation masks is still a limitation and further optimizations are in future directions

**Justification Of The Preliminary Rating:**

The paper has its merit in addressing the challenge of irregularly shaped structures for ViTs in the field of medical image computing, though with potential pitfalls and warranting further investigation.

**Questions To Address In The Rebuttal:**

Following the previous thoughts, it would be great if the authors would discuss the need for segmentation masks and whether this could be optimized or automated, and a comparison with other closely-related works.

**Special Issue:**

Yes

---

> ### Author Response · Authors · 2024-03-17
>
> Thank you for the detailed feedback on our paper. We appreciate the acknowledgement of the strengths of our proposed _VariViT_ model in addressing challenges in medical image computing, particularly in the context of irregularly shaped structures like tumors. Additionally, we thank the reviewer for recognizing the intuitiveness and effectiveness of our _center and select_ method for resizing positional embeddings and the efficiency of our proposed _CBS_ and _GA_ batching strategies.
>
> ### 1. Segmentation Mask:
>
> We recognize the importance of segmentation masks for tasks involving tumor bounding boxes, as they assist in delineating tumors and calculating their volume. This data is vital for organizing images into suitable size categories. The process necessitates prior knowledge of the Region of Interest's (ROI) size. While we understand the concern about the requirement for segmentation masks, it's important to note that these masks are essential for accurate classification and cannot be omitted. We will explore potential optimizations or automation of tumor delineation in future iterations of our paper. Further, we have added this to the discussion section of the revised report.
>
> ### 2. Multi-Class Classification:
>
> Regarding the limitation of our paper to binary classification tasks, we acknowledge the importance of addressing the evaluation of our model to multi-class tasks. To address this concern, we have extended the glioma dataset to include multi-class labels (tumor type as per _WHO2021_ classification of brain tumors [1]). We demonstrate that our proposed method maintains comparable performance to the baseline models, even in the context of multi-class classification, while exhibiting faster training times. This extension not only broadens the applicability of our approach but also highlights its versatility across various classification scenarios.
>
> ### 3. Comparison to Other SOTA Methods:
>
> The reviewer raised the issue of limited comparative analysis with other related works. We recognize the significance of such comparisons in providing a comprehensive understanding of state-of-the-art methodologies. To rectify this, we conducted additional experiments comparing our method not only with vanilla _ViT_ and _ResNet_ but also with _Pix2Struct_. While _Pix2Struct_ initially operates in a 2D domain, we extended its application to a 3D framework. Our model demonstrates better performance over _Pix2Struct_, achieving a **1.3%** improvement in the F1-score for _IDH_ status classification and a **4.3%** improvement for tumor type classification. Notably, this is achieved with up to an **8%** reduction in training time. To ensure full reproducibility, we also release our 3D implementation of _Pix2Struct_ along with the _VarViT_ code publicly at https://github.com/Aswathi-Varma/varivit. We were unable to include _NaViT_ in our comparison due to two factors. First, the authors did not release their code publicly. Second, certain aspects of their paper lacked clarity, making it difficult to implement _NaViT_ within the limited time available for rebuttal.
>
> [1] Louis, David N., et al. "The 2021 WHO classification of tumors of the central nervous system: a summary." Neuro-oncology 23.8 (2021): 1231-1251.

---

### Meta-Review · Area_Chair_CQfm · 2024-04-04

**Recommendation:** Accept (Poster)
**Confidence:** 4

**Metareview:**

This paper proposes strategies for handling variable image sizes in a ViT model.

Strengths:
+ Handling variable size images is an interesting problem in medical imaging and useful for the community
+ Proposed strategies (flexible positional embedding, alternate batching strategy) could be applied to other ViT architectures
+ Paper is clearly written

Weaknesses:
- Experimental comparisons are largely made to simple baselines (added 1 sota comparison with rebuttal)
- The added multi-class experiment shows slightly weaker or comparable performance to baseline
- Model cannot handle test image larger than that seen in training

Following rebuttal the paper, the authors included significant enhancements (new comparisons, clarifications/discussion) and the reviewers then all tended toward weak/strong accept. While there is some question regarding the experimental settings and performance, I think the the paper topic and presented strategies would be of high interest to the MIDL community, and therefore recommend accept.

---

### Decision · Program_Chairs · 2024-04-06

Accept (Poster)